# Engineering spectral properties of non-interacting lattice Hamiltonians

Ali G. Moghaddam [1,2,3*], Dmitry Chernyavsky [1], Corentin Morice [4], Jasper van Wezel [4], Jeroen van den Brink [1,5],

**1** Institute for Theoretical Solid State Physics, IFW Dresden, Helmholtzstr. 20, 01069 Dresden, Germany
**2** Department of Physics, Institute for Advanced Studies in Basic Sciences (IASBS), Zanjan 45137-66731, Iran
**3** Research Center for Basic Sciences & Modern Technologies (RBST), Institute for Advanced Studies in Basic Science (IASBS), Zanjan 45137-66731, Iran
**4** Institute for Theoretical Physics and Delta Institute for Theoretical Physics, University of Amsterdam, 1090 GL Amsterdam, The Netherlands
**5** Institute for Theoretical Physics and Würzburg-Dresden Cluster of Excellence ct.qmat, Technische Universität Dresden, 01069 Dresden, Germany
* agorbanz@iasbs.ac.ir

April 28, 2021

## Abstract

We investigate the spectral properties of one-dimensional lattices with position-dependent hopping amplitudes and on-site potentials that are smooth bounded functions of position. We find an exact integral form for the density of states (DOS) in the limit of an infinite number of sites, which we derive using a mixed Bloch-Wannier basis consisting of piecewise Wannier functions. Next, we provide an exact solution for the inverse problem of constructing the position-dependence of hopping in a lattice model yielding a given DOS. We confirm analytic results by comparing them to numerics obtained by exact diagonalization for various incarnations of position-dependent hoppings and on-site potentials. Finally, we generalize the DOS integral form to multi-orbital tight-binding models with longer-range hoppings and in higher dimensions.

## 1 Introduction

The density of states (DOS) is a key physical quantity in condensed matter physics – a plethora of electronic properties of solids depend on it, such as conductivities, thermoelectric coefficients, and screening effects [1]. In a broader context, the DOS also shows up in other areas of physics, such as optics, electronics, acoustics, and in fact for any system to which a spectrum can be assigned. Mathematically speaking, all these systems are described by Hermitian operators or matrices, which are typically large or infinite-dimensional. Their spectral features provide the most essential information about the dynamics of the systems, and in particular the energy dependence of their response functions [2]. Therefore, the ability to tailor the DOS in condensed matter systems as well as optical and photonic metamaterials is of practical interest and importance to design specific functionalities. In recent years, driven by extensive progress in fabricating photonic and acoustic metamaterials, it is more feasible than ever to manipulate and design the DOS and spectral features of these systems [3,4]. On the condensed matter side, for example, the groundbreaking fabrications of Moiré superlattices in van der Waals heterostructures and two-dimensional (2D) materials have provided a new class of electronic systems with extremely tunable low-energy bands which can host a variety of exotic, strongly correlated, and topological phenomena [5,6].

Almost all existing approaches to control or design spectral properties employ periodic structures or superlattices, for which well-established theories such as the envelope-function approximation exist [7]. In such systems, including semiconductor superlattices and all kinds of existing metamaterials, the periodicity allows the use of concepts like quasi-momenta $k$, reciprocal space, and the Brillouin zone [8]. In particular, periodicity guarantees the presence of dispersion relations $\omega(\mathbf{k})$ that simplify the theoretical description as compared to nonperiodic cases. Nevertheless, electronic bands also develop in the absence of spatial periodicity and translational invariance, due to the hybridization of neighboring atomic orbitals, as has been widely discussed in the literature [9,10]. This includes amorphous materials and quasicrystalline structures which have been shown to possess well-defined electronic bands and spectra [11]. On the other hand, one may also think of general noninteracting lattice models in which the hopping integrals and on-site potentials vary with position. Such spatial variations can be a result of locally engineering the chemical structure, position-dependent doping, or the application of external fields and perturbations. The resulting Hamiltonians lack the periodicity of the underlying lattices, and calculation of the DOS even in the non-interacting

limit becomes challenging since the quasimomentum $k$ is no longer a good quantum number.

In this paper, we explore the spectral properties of lattice models without translation symmetries, in which the parameters of the governing Hamiltonian or dynamics (such as their hopping strengths or on-site potentials) are position-dependent and vary smoothly. We construct a general approximate scheme for calculating their DOS, which becomes exact in the limit of infinitely large lattices. The starting point in this scheme is to use a mixed representation of Bloch and Wannier functions. The basis functions in this representation are defined such that at short scales, they are delocalized similarly to Bloch functions but on longer scales they appear confined to a finite extent, covering a region over which the Hamiltonian parameters vary a little (see Fig. 1(a)). We may recall that Wannier and Bloch functions, in their standard definitions, are maximally localized in real and momentum space, respectively [12,13]. The essential advantage of using partial Wannier functions is that they can be tuned such that the lattice model Hamiltonian in their basis becomes almost diagonal with small corrections which can be treated in a perturbative manner.

We will focus on tight-binding models, although our findings can be applied to all other position-dependent lattice models as long as they can be mapped to some non-interacting Hamiltonian. This includes photonic or acoustic metamaterials, as well as spin models which effectively map to a non-interacting problem, and even a mean-field superconducting Hamiltonian with smooth spatial variations in pairing potential. Recently, such position-dependent lattice models have also been discussed in the context of modeling curved spacetimes [14] to provide gravitational analogies in quantum condensed matter systems [15].

In what follows, we first introduce a basic position-dependent lattice model and our main results for its DOS in Sec. 2. We then introduce the piecewise Wannierization approach for a 1D tight-binding (TB) model with position-dependent parameters in Sec. 3. By applying perturbation theory, we derive the general formula for the DOS in Sec. 4. Then, in Sec. 5, we consider the inverse problem of constructing a lattice model that yields a given DOS. Subsequently, we elaborate on specific examples and compare the results of the perturbative scheme based on Wannier functions with numerical calculations in Sec. 6. Generalization to higher dimensions and more general TB models are provided in 7, which is followed by the conclusions in Sec. 8.

## 2 Main result

To illustrate our main result, we consider the basic example of a finite one-dimensional lattice with hopping amplitudes $t(x)$ between neighbors and on-site potentials $\mu(x)$, both being smooth and bounded functions of position $x$, with $0 < x \leq 1$. We take the lattice to consist of $\mathcal{N}$ sites at points $x_n = n/\mathcal{N}$ with $n = 1, \cdots, \mathcal{N}$. This tight-binding model corresponds to the Hamiltonian

$$\mathcal{H} = \sum_{n=1}^{\mathcal{N}} \mu(x_n) |n\rangle\langle n| + \sum_{n=1}^{\mathcal{N}-1} \left[ t(x_n) \left( |n\rangle\langle n+1| + |n+1\rangle\langle n| \right) \right], \tag{1}$$

in which $|n\rangle$ denotes a real-space localized ket. This Hamiltonian corresponds to a symmetric tridiagonal $\mathcal{N} \times \mathcal{N}$ matrix with diagonal elements $\mu(x_n)$ and off-diagonal matrix elements $t(x_n)$ for which we wish to determine the density of states $D(\omega)$ in the limit $\mathcal{N} \to \infty$. We

will prove that in this case

$$D(\omega) \;=\; \Re \int_0^1 dx \left\{ 4\,t(x)^2 - \left[ \omega - \mu(x) \right]^2 \right\}^{-1/2}, \tag{2}$$

where $\Re$ denotes the real part of the integral. We will also show in Sec. 7 the generalization of the DOS relation for more general lattice models in which we have the next-nearest-neighbor or further hoppings and more than one orbital per site. Although the form of the integrand in the DOS relation change for these cases, we show that an integral form for the DOS always exists and can be evaluated at least numerically. We should mention that there is no constraint on the functions $t(x)$ and $\mu(x)$, other than boundedness and piecewise continuity and smoothness of their variations on the lattice scale. The boundedness criterion is related to the fact that for any physically reasonable lattice model, the parameters in the Hamiltonian should be finite.

## 3   Piecewise Wannierization

In the simple case of uniform and periodic TB models, the momentum eigenstates

$$|k\rangle = \mathcal{N}^{-\frac{1}{2}} \sum_{n=1}^{\mathcal{N}} e^{ik\,n} |n\rangle \;, \tag{3}$$

diagonalize the TB Hamiltonian. It should be noticed that for a finite lattice, $k$ takes the discrete values $k_l = 2\pi l/\mathcal{N}$ with $l = 1, \cdots, \mathcal{N}$ defining the first Brillouin zone (BZ). For the nonuniform models considered here, due to the lack of translational invariance, momentum is not a good quantum number. Nevertheless, we can exploit a mixed real/momentum-space basis, which we call the basis of *partial Wannier functions* (PWFs), to approximately diagonalize the Hamiltonian. There are off-diagonal correction terms that will be shown to vanish for infinitely large lattices, and therefore can be treated in a perturbative manner for finite lattices. Figure 1(a) schematically shows the difference between full Bloch wavefunctions, maximally localized Wannier states, and the PWFs considered here.

We assume that the full TB chain consists of $M_c$ smaller chains each containing $N_s$ sites such that $\mathcal{N} = M_c N_s$ gives the number of total sites in the full lattice. Each site $n$ can be alternatively labelled with $m_c$ and $n_s$ corresponding to the position of the small chain and the place of the site inside that chain, respectively. This way, we have $n = N_s\,m_c + n_s$ and we can define the notation $|n\rangle \equiv |m_c, n_s\rangle$. The PWFs are now defined as

$$|m_c, \vartheta\rangle = N_s^{-\frac{1}{2}} \sum_{n_s=1}^{N_s} e^{i\vartheta n_s} |m_c, n_s\rangle \tag{4}$$

in which the quasi-momentum $\vartheta$ can take $N_s$ different values $\vartheta_l = 2\pi l/N_s$ with $l = 1, \cdots, N_s$. As illustrated in Fig. 1(b), this approach fictitiously divides the full lattice into smaller pieces. The PWFs are localized to only a single small chain within the full 1D lattice but within that small chain, they have an extended Bloch form, which leads us to call this approach partial or piecewise Wannierization. The full Hamiltonian likewise is divided into two types of terms, corresponding to whether they only couple the PWFs inside each small chain or couple states

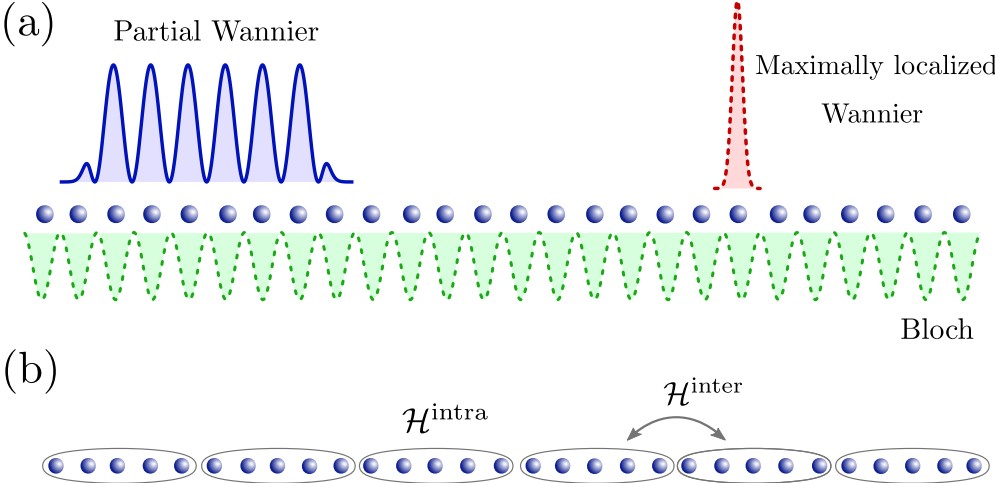

Figure 1: (Color online) (a) Illustration of Bloch functions, maximally-localized Wannier functions, and partial Wannier functions for a 1D lattice model. Partial Wannier states form a mixed basis with states that lie between Bloch states and maximally localized Wannier states. In a coarse-grained picture and on long length scales, the partial Wannier state is localized, but on smaller scales, it appears extended. (b) The division of a full lattice into smaller pieces by which the Hamiltonian can be decomposed to *intra-* and *inter*-chain parts.

between neighboring chains (These terms are labeled "intra" and "inter", respectively). It should be mentioned that if $\mathcal{H}$ includes hopping to the $z^{\text{th}}$-nearest neighbor, we should ensure $N_s > z$, so that there is only coupling between nearest-neighbor chains.

Equipped with the new basis based on PWFs, we can rewrite the Hamiltonian (1) which is readily decomposed into three parts

$$\mathcal{H} = \mathcal{H}_0^{\text{intra}} + \mathcal{H}_1^{\text{intra}} + \mathcal{H}_1^{\text{inter}} . \tag{5}$$

The three terms above correspond to the diagonal terms in the PWF basis, block-diagonal corrections to the intra-chain parts of the Hamiltonian due to the spatial variation of $t(x)$ and $\mu(x)$, and block-off-diagonal coupling terms originating from inter-chain hopping terms, respectively. We will see that the two correction terms $\mathcal{H}_1^{\text{intra}}$ and $\mathcal{H}_1^{\text{inter}}$ respectively scale as $\mathcal{O}(M_c^{-1})$ and $\mathcal{O}(N_s^{-1})$ which implies that by assuming both large $M_c$ and large $N_s$, they can be treated as small perturbations.

On the other hand, the nonvanishing matrix elements in the PWF basis can be categorized in three different groups:

- $\langle m_c, \vartheta' | \mathcal{H}_{\text{hop}} | m_c, \vartheta \rangle$: the hopping contributions to $\mathcal{H}_0^{\text{intra}}$ and $\mathcal{H}_1^{\text{intra}}$;

- $\langle m_c, \vartheta' | \mathcal{H}_{\text{onsite}} | m_c, \vartheta \rangle$: the onsite potential contributions to $\mathcal{H}_0^{\text{intra}}$ and $\mathcal{H}_1^{\text{intra}}$;

- $\langle m_c, \vartheta' | \mathcal{H}_{\text{hop}} | m_c \pm 1, \vartheta \rangle$: the inter-chain part of the Hamiltonian, $\mathcal{H}_1^{\text{inter}}$.

The matrix elements $\langle m_c, \vartheta' | \mathcal{H}_{\mathrm{hop}} | m_c, \vartheta \rangle$ are evaluated as

$$
\begin{aligned}
&\langle m_c, \vartheta' | \mathcal{H}_{\mathrm{hop}} | m_c, \vartheta \rangle \\
&= \frac{1}{N_s} \sum_{n'_s=1}^{N_s} \sum_{n_s=1}^{N_s} \sum_{n=1}^{\mathcal{N}} e^{i\vartheta n_s - i\vartheta' n'_s} \, t\left(\frac{n}{\mathcal{N}}\right) \langle m_c, n'_s | \left( |n\rangle\langle n+1| + |n+1\rangle\langle n| \right) |m_c, n_s\rangle \\
&= \frac{1}{N_s} \sum_{n'_s=1}^{N_s} \sum_{n_s=1}^{N_s} e^{i\vartheta n_s - i\vartheta' n'_s} \left[ t\left(\frac{N_s\, m_c + n'_s}{\mathcal{N}}\right) \delta_{n_s, n'_s+1} + t\left(\frac{N_s\, m_c + n_s}{\mathcal{N}}\right) \delta_{n_s+1, n'_s} \right] \\
&= \frac{\left(e^{i\vartheta} + e^{-i\vartheta'}\right)}{N_s} \sum_{n_s=1}^{N_s} e^{i(\vartheta - \vartheta')n_s} \, t\left(\frac{N_s\, m_c + n_s}{\mathcal{N}}\right).
\end{aligned}
\tag{6}
$$

Since $n_s/\mathcal{N} \leq 1/M_c \ll 1$, we can Taylor-expand the hopping term up to first order which results in

$$
\begin{aligned}
\langle m_c, \vartheta' | \mathcal{H}_{\mathrm{hop}} | m_c, \vartheta \rangle \;\approx\;& t\left(\frac{m_c}{M_c}\right) 2\cos\vartheta \, \delta_{\vartheta\,\vartheta'} \\
+\;& t'\left(\frac{m_c}{M_c}\right) \frac{\left(e^{i\vartheta} + e^{-i\vartheta'}\right)}{M_c} \sum_{n_s=1}^{N_s} \frac{n_s}{N_s^2} e^{i(\vartheta-\vartheta')n_s} + \mathcal{O}\left(\frac{1}{M_c^2}\right),
\end{aligned}
\tag{7}
$$

with $t'(x)$ denoting the derivative of $t(x)$ evaluated at $x$. The first term in Eq. (7) is diagonal and comes from approximating the Hamiltonian of this portion of the full chain with a system of uniform hopping $t(m_c/M_c)$. The second term in the right-hand side of Eq. (7) consists of both diagonal and off-diagonal terms in general, which are both of order $1/M_c$.

In a similar way as above, the matrix elements of the onsite potential terms in the PWF basis can be evaluated, and read

$$
\begin{aligned}
\langle m_c, \vartheta' | \mathcal{H}_{\mathrm{onsite}} | m_c, \vartheta \rangle \;=\;& \frac{1}{N_s} \sum_{n'_s=1}^{N_s} \sum_{n_s=1}^{N_s} \sum_{n=1}^{\mathcal{N}} e^{i\vartheta n_s - i\vartheta' n'_s} \, \mu\left(\frac{n}{\mathcal{N}}\right) \langle m_c, n'_s | n\rangle\langle n | m_c, n_s\rangle \\
=\;& \frac{1}{N_s} \sum_{n_s=1}^{N_s} e^{i(\vartheta - \vartheta')n_s} \, \mu\left(\frac{N_s m_c + n_s}{\mathcal{N}}\right) \\
=\;& \mu\left(\frac{m_c}{M_c}\right) \delta_{\vartheta\,\vartheta'} + \mu'\left(\frac{m_c}{M_c}\right) \frac{1}{M_c} \sum_{n_s=1}^{N_s} \frac{n_s}{N_s^2} e^{i(\vartheta-\vartheta')n_s} + \mathcal{O}\left(\frac{1}{M_c^2}\right).
\end{aligned}
\tag{8}
$$

Collecting corresponding terms from Eqs. (7) and (8) and dropping higher-order terms which are $\mathcal{O}\left(M_c^{-2}\right)$, we arrive at the intra-chain parts of the Hamiltonian:

$$
\mathcal{H}_0^{\mathrm{intra}} = \sum_{m_c, \vartheta} \left[ t\left(\frac{m_c}{M_c}\right) 2\cos\vartheta + \mu\left(\frac{m_c}{M_c}\right) \right] |m_c, \vartheta\rangle\langle m_c, \vartheta|,
\tag{9}
$$

$$
\mathcal{H}_1^{\mathrm{intra}} = \frac{1}{M_c} \sum_{m_c, \vartheta, \vartheta'} \left[ t'\left(\frac{m_c}{M_c}\right) \left(e^{i\vartheta} + e^{-i\vartheta'}\right) + \mu'\left(\frac{m_c}{M_c}\right) \right] \mathcal{B}(\vartheta - \vartheta') |m_c, \vartheta\rangle\langle m_c, \vartheta'|,
\tag{10}
$$

with $\mathcal{B}(x) = N_s^{-2} \sum_{n_s=1}^{N_s} e^{i\,x\,n_s}$.

Because there is also hopping between the sites from the neighboring small chains, we should consider the matrix elements $\langle m_c, \vartheta' | \mathcal{H} | m_c + 1, \vartheta \rangle$ and their conjugates $\langle m_c + 1, \vartheta' | \mathcal{H} | m_c, \vartheta \rangle$. They can be similarly evaluated as

$$
\begin{aligned}
\langle m_c, \vartheta' | \mathcal{H} | m_c + 1, \vartheta \rangle &= \frac{1}{N_s} \sum_{n'_s=1}^{N_s} \sum_{n_s=1}^{N_s} \sum_{n=1}^{\mathcal{N}} e^{i\vartheta n_s - i\vartheta' n'_s} \, t\left(\frac{n}{\mathcal{N}}\right) \\
&\quad \times \langle m_c, n'_s | \left( |n\rangle\langle n+1| + |n+1\rangle\langle n| \right) | m_c + 1, n_s \rangle \\
&= \frac{1}{N_s} e^{i\vartheta - i\vartheta' N_s} \, t\left(\frac{N_s m_c + N_s}{\mathcal{N}}\right) = \frac{1}{N_s} e^{i\vartheta} \, t\left(\frac{m_c + 1}{M_c}\right) ,
\end{aligned}
\tag{11}
$$

by noticing $\vartheta' N_s = 2\pi l'$ with $l'$ being an integer inside the range of $[0, N_s]$. Assuming a large number of sites inside each small chain, the inter-chain coupling terms can again be treated perturbatively, since they are of order $1/N_s$ . Then, the inter-chain contribution of the total Hamiltonian reads

$$
\mathcal{H}_1^{\text{inter}} = \frac{1}{N_s} \sum_{m_c, \vartheta, \vartheta'} t\left(\frac{m_c + 1}{M_c}\right) \left( e^{i\vartheta} | m_c + 1, \vartheta \rangle \langle m_c, \vartheta' | + e^{-i\vartheta'} | m_c, \vartheta \rangle \langle m_c + 1, \vartheta' | \right).
\tag{12}
$$

# 4  Perturbation theory based on PWFs

Now that we calculated the terms of the Hamiltonian (5) in the PWF basis, we can apply perturbation theory to this Hamiltonian. Since we already know the matrix elements of all terms in the Hamiltonian, the zeroth and first-order energies can be simply obtained as

$$
E_{m_c, \vartheta}^{(0), \text{intra}} = \mu\left(\frac{m_c}{M_c}\right) + 2\, t\left(\frac{m_c}{M_c}\right) \cos\vartheta,
\tag{13}
$$

$$
E_{m_c, \vartheta}^{(1), \text{intra}} = \frac{1}{M_c} \langle m_c, \vartheta | \mathcal{H}_1^{\text{intra}} | m_c, \vartheta \rangle = \frac{1}{M_c} \left[ \frac{1}{2} \mu'\left(\frac{m_c}{M_c}\right) + t'\left(\frac{m_c}{M_c}\right) \cos\vartheta \right],
\tag{14}
$$

$$
E_{m_c, \vartheta}^{(1), \text{inter}} = 0.
\tag{15}
$$

The first-order correction due to the intra-chain part of the Hamiltonian is nothing but the linear correction to the position-dependence of the hoppings inside the chain, whereas the inter-chain part which is block-off-diagonal has no first-order contribution. We can calculate the second-order correction due to $\mathcal{H}_1^{\text{inter}}$ which reads

$$
\begin{aligned}
E_{m_c, \vartheta}^{(2), \text{inter}} &= \frac{1}{N_s^2} \sideset{}{'}\sum_{m'_c, \vartheta'} \frac{\left| \langle m_c, \vartheta | \mathcal{H}_1^{\text{inter}} | m'_c, \vartheta' \rangle \right|^2}{E_{m_c, \vartheta}^{(0)} - E_{m'_c, \vartheta'}^{(0)}} \\
&= \frac{1}{2 N_s^2} \sum_{\vartheta'} \left[ \frac{\left| t\left(\frac{m_c+1}{M_c}\right) \right|^2}{t\left(\frac{m_c}{M_c}\right) \cos\vartheta - t\left(\frac{m_c+1}{M_c}\right) \cos\vartheta'} \right. \\
&\qquad\qquad\qquad \left. + \frac{\left| t\left(\frac{m_c}{M_c}\right) \right|^2}{t\left(\frac{m_c}{M_c}\right) \cos\vartheta - t\left(\frac{m_c-1}{M_c}\right) \cos\vartheta'} \right].
\end{aligned}
\tag{16}
$$

We note that the double sum in the first line above excludes the single term with $m'_c = m_c$ and $\vartheta' = \vartheta$. The single sum in the second line however, covers all possible values of $\vartheta'$, including $\vartheta$,

owing to the fact that we have only nonvanishing matrix elements for $m_c' = m_c \pm 1$. If $\cos \vartheta = 0$, we can immediately see that $E_{m_c,\vartheta}^{(2),\,\text{inter}}$ vanishes since it is proportional to $\sum_{\vartheta'} 1/\cos \vartheta' \equiv 0$. Otherwise, assuming $\cos \vartheta \neq 0$, we arrive at the approximate expression

$$E_{m_c,\vartheta}^{(2),\,\text{inter}} \approx \frac{1}{N_s^2} \left\{ \left[ t\left(\frac{m_c}{M_c}\right) + \frac{1}{M_c} t'\left(\frac{m_c}{M_c}\right) \right] \sum_{\vartheta' \neq \vartheta} \frac{1}{\cos \vartheta - \cos \vartheta'} - t\left(\frac{m_c}{M_c}\right) \frac{1}{\cos \vartheta} \right\}, \qquad (17)$$

which is a term of the order of $\mathcal{O}(N_s^{-2})$. In the limit of $M_c, N_s \gg 1$, the energy spectrum of the position-dependent TB model can thus be well approximated by the zeroth-order energy expectation values of the PWFs. For $\mathcal{N} \to \infty$ we can safely consider both $M_c$ and $N_s$ going to infinity as well, and the approximations we made become asymptotically exact.

As an immediate implication of the perturbative scheme based on PWFs, the DOS per site can be written as

$$
\begin{aligned}
D(\omega) &\approx \frac{1}{\mathcal{N}} \sum_{m_c,\vartheta} \delta\left(\omega - E_{m_c,\vartheta}^{(0)}\right) = \frac{1}{M_c} \sum_{m_c=1}^{M_c} \frac{1}{N_s} \sum_{l=0}^{N_s-1} \delta\left[\omega - \mu\left(\frac{m_c}{M_c}\right) - 2\,t\left(\frac{m_c}{M_c}\right) \cos \vartheta_l\right] \\
&= \int_0^1 dx \int_0^{2\pi} \frac{d\vartheta}{2\pi} \, \delta\left[\omega - \mu(x) - 2\,t(x) \cos \vartheta\right]
\end{aligned}
\qquad (18)
$$

using only the lowest-order energies $E_{m_c,\vartheta}^{(0)}$. By performing the integration over the quasi-momentum $\vartheta$, we acquire the DOS relation of Eq. (2). Recall that $\vartheta_l = 2\pi l/N_s$ for $l \in \mathbb{Z}_{N_s}$, and that the integrals in the second line originates from taking the limits $N_s \to \infty$ and $M_c \to \infty$ of the two summations, respectively. As mentioned before, this implies that for infinitely large lattices and assuming smooth spatial variations of Hamiltonian parameters such as $t(x)$ and $\mu(x)$, the zeroth-order energy expression $E_{m_c,\vartheta}^{(0)}$ and the DOS relation of Eq. (2) become exact. In Sec. 5, we will compare the DOS profiles obtained from the above analytic relation with those of numerical diagonalization for various finite-sized position-dependent models, to demonstrate the versatile applicability of expression (2).

# 5    Constructing lattice model to match a given DOS

We now consider the inverse problem of finding a lattice model which gives a prescribed DOS. In contrast to the more common cases where we know the Hamiltonian, here the Hamiltonian is unknown and will be derived based on knowledge about the DOS. Such inverse problems have already been explored in the mathematics community, yet there are theoretic challenges about the necessary or sufficient conditions for the existence and uniqueness of solutions, and also practical questions about suitable algorithms and numerical methods [16]. As a matter of fact, infinitely many different operators can have the same spectrum and therefore equal DOS. However, we can refine this situation by concentrating on the special family of lattice models with position-dependent hopping and/or onsite-potentials as introduced earlier. We concentrate here on the case of an infinite lattice with just nearest-neighbor hopping and in the presence of the particle-hole symmetry condition $\mu(x) = 0$. We then proceed to calculate $t(x)$ such that Eq. (2) yields a desired DOS, meaning that we consider the DOS relation as an integral equation. In general, even if there exists a set of $t(x)$ that matches a given $D(\omega)$, finding them as a solution of the integral equation is only possible numerically. However, for

the particle-hole symmetric cases, a formal analytical solution exists, since we can transform Eq. (2) to,

$$D(\omega) = \Re \int_{t(0)}^{t(1)} dt \, \mathcal{K}(\omega, t) \, \mathcal{Y}(t) = \int_{|\omega|/2}^{t_{\max}} dt \, \mathcal{K}(\omega, t) \, \mathcal{Y}(t), \tag{19}$$

$$\mathcal{K}(\omega, t) = \left(4t^2 - \omega^2\right)^{-1/2}, \tag{20}$$

with $\mathcal{Y}(t) = (dt/dx)^{-1}$ being the inverse of the derivative of the hopping parameter $t(x)$ rewritten as a function of $t$. The expression (19) assuming a given $D(\omega)$ defines a Volterra integral equation of the first kind. For the special kernel $\mathcal{K}(\omega, t)$ arising here, we show in Appendix A that a formal solution

$$\mathcal{Y}(t) = \frac{-1}{\pi} \frac{d}{dt} \int_{2t}^{\omega_{\max}} \omega \, d\omega \, \frac{D(\omega)}{\sqrt{\omega^2 - 4t^2}} \tag{21}$$

exists, where the upper bound $\omega_{\max}$ is dictated by the bandwidth above which the DOS vanishes. Note that using the particle-hole symmetry condition $D(\omega) = D(-\omega)$, we can replace the integration over all energies with twice the integral over just positive energies. Having $\mathcal{Y}(t)$ in hand, we define

$$\chi(t) = \int_0^t dt' \, \mathcal{Y}(t') = \frac{1}{\pi} \left[ \int_0^{\omega_{\max}} d\omega \, D(\omega) - \int_{2t}^{\omega_{\max}} \omega \, d\omega \, \frac{D(\omega)}{\sqrt{\omega^2 - 4t^2}} \right] = x, \tag{22}$$

whose inverse function gives the spatial form of the hopping as $t(x) = \chi^{-1}(x)$.

The only limitations to devising a 1D TB model with position-dependent hoppings giving a desired particle-hole symmetric DOS are the integrability criteria for Eqs. (21) and (22). Although it is not necessary, having a bounded smooth function $D(\omega)$ provides a sufficient condition for the convergence of both integrals. Therefore by varying $t(x)$ almost any smooth DOS can be obtained.

A case of particular interest is when the DOS has singular behavior at particular energies, a special feature known as a van Hove singularity. Simple 1D TB models typically have van Hove singularities at the band edges as $D(\omega) \propto \sqrt{\omega_{\max}^2 - \omega^2}$, yet it remains finite and differentiable, otherwise. Interestingly, for the position-dependent TB model, we observe that it is possible to generate a DOS with any singularity of the form $\omega^{-\beta}$ at the center of the band as long as $\beta < 1$. But higher-order singularities with $\beta \geq 1$ cannot occur in the nearest-neighbor hopping models as the integral in (21) then becomes divergent. The significance of Van Hove singularities has been noticed long ago, as in their presence the sensitivity of the system to perturbations and interactions is substantially enhanced. The notion of higher-order van Hove singularities, with power-law form $D(\omega) \propto \omega^{-\beta}$ opposed to the logarithmic behavior at an ordinary Van Hove singularity, has also recently been put forward in the context of two-dimensional materials, and particularly Moiré superlattices such as twisted bilayer graphene [17–19].

In general, as long as there exists a solution, one can perform the integration in (22) numerically to obtain $t(x)$ which yields a given DOS. But in certain situations, analytical closed forms for $t(x)$ can be obtained starting from a given form of $D(\omega)$. So, to elucidate the mathematical procedure of obtaining hopping parameters $t(x)$, we explicitly examine the two interesting examples of a constant ($D_1(\omega)$) and a semicircular ($D_2(\omega)$) DOS with bandwidth

$W = 2$. By evaluating the integration and derivative in Eq. (21) we find that

$$D_1(\omega) = 1 \qquad \Longrightarrow \qquad \mathcal{Y}_1(t) = \frac{dx}{dt} = \frac{4t}{\pi\sqrt{1-4t^2}} \tag{23}$$

$$D_2(\omega) = \sqrt{1-\omega^2} \qquad \Longrightarrow \qquad \mathcal{Y}_2(t) = \frac{dx}{dt} = 2t \tag{24}$$

where the solutions for $t(x)$ assuming $t(x=0) = 0$ are respectively

$$t_1(x) = \sqrt{x\,(1-x)}, \qquad 0 \le x \le 1, \tag{25}$$
$$t_2(x) = \sqrt{x}, \qquad 0 \le x \le 1. \tag{26}$$
$$\tag{27}$$

It is interesting to note that the case of constant DOS is equivalent to having equally-distanced eigenenergies throughout a bounded spectrum and this has been studied for finite tridiagonal matrices [20–22]. It has been found that a tridiagonal symmetric matrix of order $n$ with off-diagonal entries

$$a_{j,j+1} = \frac{\sqrt{j(2n-j-1)}}{2} \quad (j = 1, \cdots, n-2)\,, \qquad a_{n-1,n} = \sqrt{\frac{n(n-1)}{2}}, \tag{28}$$

has equally-distanced eigenvalues which match the form of the hopping parameter $t(x)$ we find here. The only exception is that the lowest off-diagonal entry $a_{n-1,n}$ has an extra prefactor of $\sqrt{2}$ compared to one expected from the trend of $a_{j,j+1}$. Such a difference can be regarded as a finite-size effect and that can be ignored for large $n$. We should note that, unlike the famous example of the quantum harmonic oscillator, which also has constant level spacing, here the range of the energies is bounded. Although a constant DOS for a finite range can be formally obtained assuming a linear dispersion relation $\varepsilon(k) \propto k$, such a dispersion is anomalous, meaning that it does not correspond to any 1D lattice model with uniform hoppings. This example suggests that, unlike the case of position-dependent lattice models, it is not *a priori* guaranteed that any uniform-hopping models exist corresponding to a prescribed DOS. We will address this problem in the following in more detail, and show how one can construct uniform-hopping models from the DOS. As a final remark on the constant DOS, we would like to mention that it has been long-known that the nonsymmetric tridiagonal matrices with entries $a_{i,i+1} = i$ and $a_{i+1,i} = n - i$ also have equally-distanced eigenvalues [23, 24]. These matrices are known as Sylvester-Kac matrices and from a physical point of view, they represent a non-Hermitian 1D position-dependent TB model.

## 5.1 Correspondence to periodic lattice models

We now consider the construction of lattice models giving a desired DOS, but for the uniform, position-independent case, in the presence of hopping between non-neighboring sites. The Hamiltonian of such uniform lattice models in 1D and for a single orbital reads

$$\mathcal{H}_{\text{uniform}} = \sum_{n,m} \xi_m \, |n\rangle\langle n+m| \tag{29}$$

where $\xi_m$ ($m \ge 1$) denotes the hopping between $m^{\text{th}}$ nearest neighboring sites and $\xi_0$ is a constant on-site potential. We will see that, by considering the hopping between distant sites

and tuning their relative strengths $\xi_m$, we can also engineer the DOS. This will serve as a kind of correspondence between the position-dependent and position-independent lattice models with short- and long-range hoppings, respectively, meaning that their DOS becomes identical in the limit of very large lattice sizes. Here, a lattice model with short-range and long-range hoppings simply refers to whether the hopping either identically vanishes when the distance between two sites becomes larger than a finite length, or the hoppings between even very distant sites remain non-zero.

For a generic single-band model with position-independent hopping such as Eq. (29), and assuming periodic boundary conditions, there exists a dispersion relation $\varepsilon(k)$. So, the DOS can be written as

$$D(\omega) = \int \frac{dk}{2\pi}\, \delta\big[\omega - \varepsilon(k)\big] = \int \frac{dk}{2\pi}\, \frac{\delta\big[k - \varepsilon^{-1}(\omega)\big]}{|d\varepsilon/dk|} = \frac{1}{|d\varepsilon/dk|_{k=\varepsilon^{-1}(\omega)}}, \tag{30}$$

where $\varepsilon(k)$ has been assumed to be an invertible function [1], although it is not a crucial constraint and can be relaxed by dividing the full range of the parameter $k$ into regions for which $\varepsilon(k)$ is monotonic and has an inverse $\varepsilon_i^{-1}(\omega)$ corresponding to the $i^{th}$ region. In the case of completely monotonic function $\varepsilon(k)$ only one $k$ exists and consequently, we can simply integrate the equation (30) as

$$k = \pm \int^{\omega} d\omega'\, D(\omega') \equiv \varepsilon^{-1}(\omega), \tag{31}$$

which gives the inverse of the dispersion relation as an integral of the DOS. So the problem reduces to finding the lattice model parameters $\xi_m$ such that the resulting dispersion relation matches with the result of Eq. (31) for a given DOS. The dispersion relation for the Hamiltonian (29) is given by

$$\varepsilon(k) = 2\sum_m \xi_m \cos(mk) \equiv \omega\ . \tag{32}$$

This is simply a Fourier series and therefore the Hamiltonian parameters $\xi_m$ can be obtained as

$$\xi_m = \frac{1}{2} \int_0^{2\pi} \frac{dk}{2\pi}\, \varepsilon(k)\, \cos(mk) = \frac{1}{2} \int_{-\omega_{\max}}^{\omega_{\max}} d\omega\, \omega\, D(\omega)\, \cos\big[m\, \varepsilon^{-1}(\omega)\big]\ , \tag{33}$$

where we have changed the integration variable from $k$ to $\omega$ in the final expression. Invoking Eq. (31), we arrive at

$$\xi_m = \int_0^{\omega_{\max}} d\omega\, \omega\, D(\omega)\, \cos\left[m \int^{\omega} d\omega'\, D(\omega')\right]\ , \tag{34}$$

which gives $\xi_m$ in terms of the DOS. In the next section, we compare examples of position-dependent hopping models to corresponding position-independent models with the same DOS. We will see that for all periodic models we consider, with a DOS coinciding with that of a position-dependent lattice model, we must include long-range hoppings between far neighbors.

---

[1] This occurs for a completely non-degenerate spectrum

# 6    Examples and comparison with numerics

Having derived analytical integral expressions, we explicitly examine some position-dependent TB models, by calculating their DOS using Eq. (2) and comparing them with direct numerical calculations of the spectrum. We focus on the case of a 1D chain with power-law variation of the hopping strength: $t_n = [n/(\mathcal{N} - 1)]^\gamma$ ($\gamma \geq 0$). Such a power-law form has been motivated previously by the fact that the resulting low-energy (long-wavelength continuum) physics corresponds to a 1D Dirac equation subjected to a gravitational background that possesses a horizon for $\gamma \geq 1$. In addition, as we will see in the following, the DOS for power-law variation with $\gamma \geq 1$ has a van Hove singularity at $\omega = 0$. In fact, the DOS for these cases has a closed-form expression given by

$$D(\omega) = \Re \int_0^1 dx \, \frac{1}{\sqrt{\left[2\alpha x^\gamma\right]^2 - \omega^2}} = \frac{-1}{|\omega|} \Im \left[ {}_2F_1 \left( \frac{1}{2}, \frac{1}{2\gamma}, 1 + \frac{1}{2\gamma}; \frac{4\alpha^2}{\omega^2} \right) \right], \tag{35}$$

where ${}_2F_1(a, b, c; z)$ denotes the hypergeometric function with three real parameters $a$, $b$, $c$, and the variable $z$. Using the limiting behavior of the hypergeometric function, we find that, for any $\gamma > 1$, the DOS is singular as $D(\omega) \propto \omega^{1/\gamma - 1}$ at zero energy ($\omega \to 0$). This result is of practical interest because it introduces a model with a divergent DOS at zero energy, in the middle of the band, whereas in ordinary 1D models with uniform hopping constants, possible divergences known as Van Hove singularities occur at the edges of the band. When $\gamma < 1$ we find, in contrast, a smooth behavior for the DOS without any singularity at $\omega = 0$. For the special case of $\gamma = 1/2$, the DOS turns out to be identical to the *Wigner semicircle distribution*

$$D_{\gamma = 1/2}(\omega) = \frac{1}{2\alpha^2} \sqrt{4\alpha^2 - \omega^2}, \tag{36}$$

as already noticed in the previous section. The Wigner semicircle distribution is known to emerge in $n \times n$ symmetric random matrices with independent and identically distributed entries in the large $n$ limit [25, 26]. Here, in contrast, we find a particular tridiagonal matrix with off-diagonal entries varying as $M_{i+1,i} = M_{i,i+1} = \sqrt{i/n}$ that yield a semicircular form for its DOS. Then, for the marginal case of $\gamma = 1$, we can also find a simple form

$$D_{\gamma = 1}(\omega) = \frac{1}{2\alpha} \log \left( \frac{2\alpha + \sqrt{4\alpha^2 - \omega^2}}{|\omega|} \right), \tag{37}$$

which has a logarithmic divergence at $\omega = 0$. As displayed in Fig. 2, we find good agreement between these results and those obtained from numerical calculations for large enough lattices and different exponents $\gamma$.

The effects of position-dependent on-site potential assuming both uniform and position-dependent hopping parameters are shown in Fig. 3 for various combinations of power-law forms of $\mu(x)$ and $t(x)$. Similar to the case of a power-law form of only the hopping, the power-law varying on-site potential can yield singularities in the DOS as shown in Figs. 3(a-d). As expected, the nonzero on-site potential breaks the electron-hole symmetry and as a result, the position and type of the singularities in the DOS can be tuned by changing $\mu(x)$. From a practical point of view, introducing a position-dependent on-site potential can be done by applying external electric fields or other perturbations.

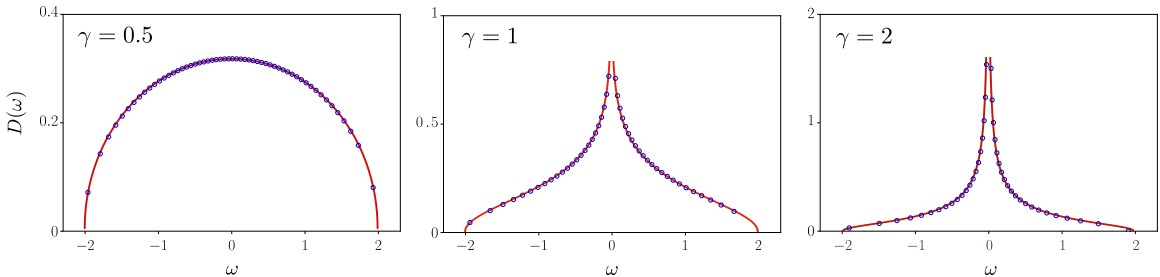

Figure 2: (Color online) DOS profiles for 1D lattice models with power-law variations of the hopping parameter $t(x) \propto x^\gamma$ for various $\gamma$. Solid red lines are obtained from the analytical relation whereas the blue circles come from numerical calculations using a lattice with $\mathcal{N} = 500$. The left panel shows the Wigner semicircular shape expected for $\gamma = 1/2$ whereas the two other cases with $\gamma \geq 1$ show the hallmark divergence of the DOS at zero energy.

Next, we provide some examples to illustrate how we find the lattice-periodic TB models with long-range hopping corresponding to a given DOS, based on Eqs. (31) and (33). Particularly, we consider the two DOS relations (36) and (37) which arise for the position-dependent hopping models with $\gamma = 1/2$ and $\gamma = 1$, respectively. Plugging these forms of the DOS into (31) we find the inverse of the dispersion-like relations to be [2]

$$k = \varepsilon^{-1}_{\gamma=1/2}(\omega) = 2\left[ \arcsin\left(\frac{\omega}{2\alpha}\right) + \frac{\omega}{W}\sqrt{1 - \left(\frac{\omega}{2\alpha}\right)^2} \right], \tag{38}$$

$$k = \varepsilon^{-1}_{\gamma=1}(\omega) = 2\arcsin\left(\frac{\omega}{2\alpha}\right) + \frac{\omega}{\alpha}\operatorname{arccosh}\left(\frac{2\alpha}{|\omega|}\right). \tag{39}$$

Then, using (33), we can obtain the $m^{\text{th}}$ order hopping strength of the periodic TB model corresponding to these dispersion relations. Interestingly, the hopping between far neighbors ($m \gg 1$) for these cases approximately behaves as $\xi_m \propto -(-1)^m/m$ for $m \gg 1$, corresponding to a long-range hopping model. The numerical results show that it is plausible that there generically exist periodic long-range hopping models and short-range position-dependent hopping models yielding the same density of states. From a mathematical point of view, such a correspondence between position-dependent short-range TB models and long-range lattice-periodic ones, indicates the existence of a similarity transformation between tridiagonal matrices and the so-called Toeplitz (or diagonal-constant) matrices [27].

Finally, we examine the deviations between numerically obtained energies using exact diagonalization of a finite lattice Hamiltonian and the analytical results of the perturbative scheme of Sec. 4. As an example, we consider the power-law varying hopping parameter $t(x) = x^2$ and show the numerical and analytical results (obtained from by (13)) in Fig. 4(a) and their difference in 4(b). The deviation, when we also include the first order corrections given by Eq. (14), is shown in 4(c). The numerical calculations are done for a lattice size $\mathcal{N} = 1600$ and for the perturbative expression, we have divided the full chain into $M_c = 40$ smaller chains each consisting of $N_s = 40$ sites. The results clearly indicate that the difference between exact numerics and the perturbative framework of PWFs is very small even for a lattice size of the order of $\mathcal{N} \sim 10^3$. Increasing the lattice size, the deviations become smaller

---

[2]Using the identity $\operatorname{arccosh}(x) = \log\left(x + \sqrt{x^2 - 1}\right)$.

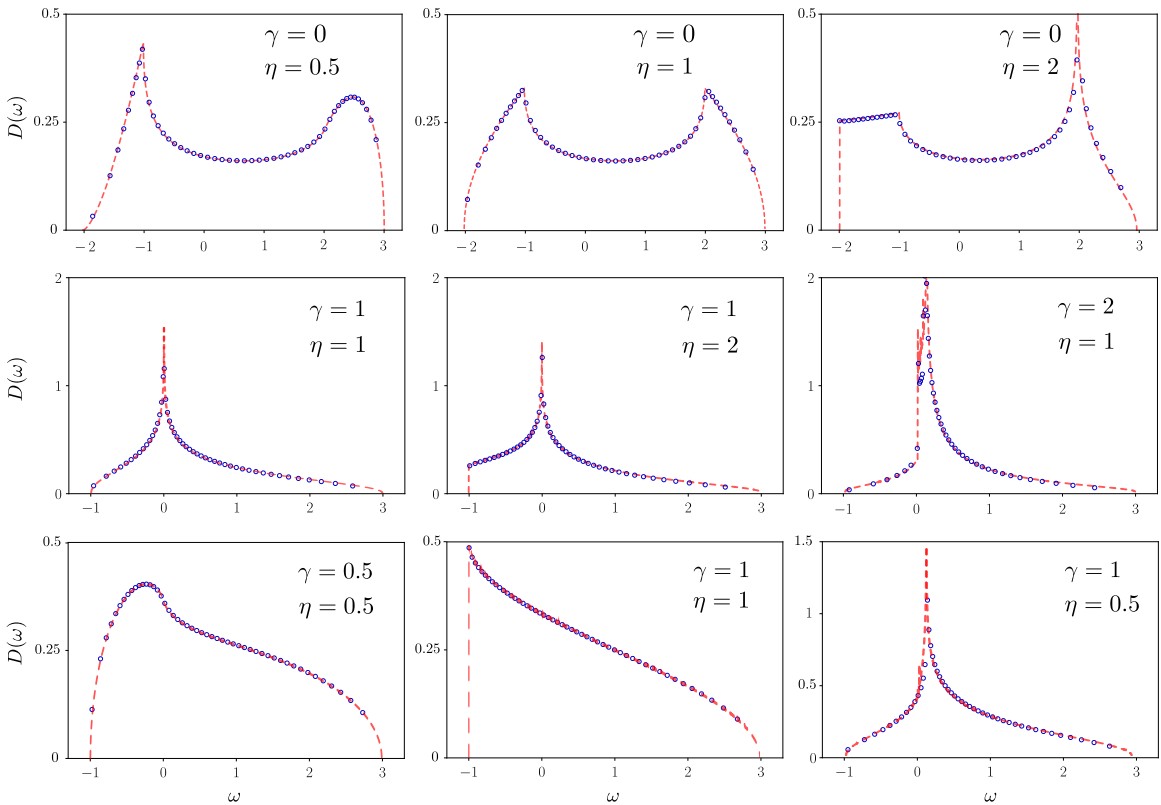

Figure 3: (Color online) The DOS profile for various forms of 1D lattice model with position-dependent hopping $t(x) \propto x^\gamma$ and on-site potential $\mu(x) \propto x^\eta$. Solid red lines are obtained from the analytical relation, whereas the blue circles come from the numerical calculations for a lattice with $\mathcal{N} = 500$. One clear signature of on-site potential is to break electron-hole symmetry of the model as is evident from the DOS plots.

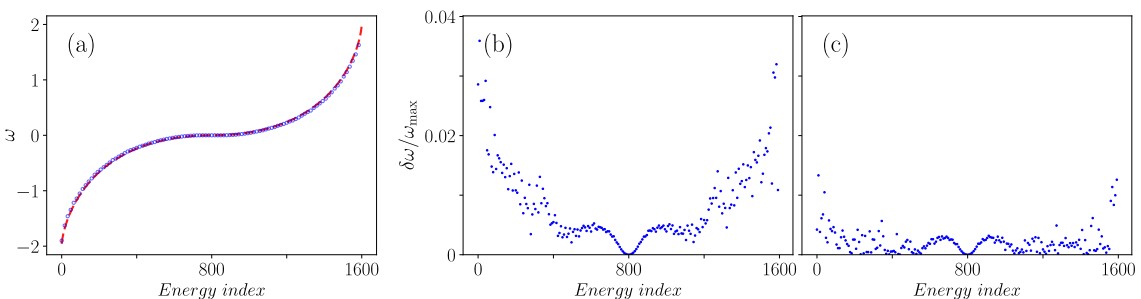

Figure 4: (Color online) The profile of energies and their differences for the values obtained from numerical and analytical calculations for the hopping parameter $t(x) \propto x^2$ and vanishing on-site potential. The lattice size is chosen to be $\mathcal{N} = 1600$ and we assume $M_c = N_s = 40$ when dividing the lattice into smaller chains. (a) Numerically (blue circles) versus analytically (red line) obtained values of energy. (b) Absolute values of the difference between numerically and analytically calculated energies to leading order in Eq. (13). (c) The difference for analytically calculated energies with the leading and the first-order perturbation terms and numerically obtained energy values.

and they vanish in the infinite-size limit.

## 7 Extension to more general lattice models

So far we have shown that the DOS of single-orbital 1D TB models with a general position-dependent nearest-neighbor hopping and on-site potential can be approximated by Eq. (2). In what follows, we explain in detail how this result can be extended for more general lattice models, such as multi-orbital (multi-band) cases, or those with farther-neighbor hoppings, and finally for higher dimensions.

First, recall that a general TB model consists of $m$ different orbitals (internal degrees of freedom) and also possibly farther-neighbor hoppings. The position-space kets can be labeled by $|\boldsymbol{i}, \eta\rangle$ where $\boldsymbol{i}$ indicates the site position and $\eta$ accounts for the different orbitals. The general TB Hamiltonian, then, can be written as

$$\mathcal{H}_{\mathrm{TB}} = \sum_{\boldsymbol{i},\boldsymbol{j}} \sum_{\eta,\eta'}^{m} t_{\boldsymbol{i},\boldsymbol{j}}^{\eta,\eta'} |\boldsymbol{i},\eta\rangle\langle\boldsymbol{j},\eta'| + \sum_{\boldsymbol{i}} \sum_{\eta,\eta'}^{m} \mu_{\boldsymbol{i}}^{\eta,\eta'} |\boldsymbol{i},\eta\rangle\langle\boldsymbol{i},\eta'|, \tag{40}$$

in which instead of a single hopping parameter we have a set of hoppings between potentially different orbitals at different sites. Similarly, different orbitals or sub-lattices corresponding to a single unit-cell can have all sorts of different on-site potentials which can be gathered in a $m \times m$ matrix. For lattice-periodic cases, the Hamiltonian is parameterized by a finite set of *position-independent* hoppings and on-site potentials, for which we use the cumulative notations $\boldsymbol{t}$ and $\boldsymbol{\mu}$. Consequently, we can use a full Bloch basis labeled as $\psi_{\mathbf{k},\ell}$ to diagonalize the Hamiltonian, which yields $m$ bands $\varepsilon_{\ell,\mathbf{k}}(\boldsymbol{t},\boldsymbol{\mu})$ in which we emphasize the implicit dependence on $\boldsymbol{t}$ and $\boldsymbol{\mu}$.

When hopping parameters and on-site potentials vary with position, we can use the gen-

eralized form of the PWF basis

$$|\boldsymbol{m}_c, \boldsymbol{\vartheta}\,;\,\tilde{\ell}\rangle = \sum_{\boldsymbol{n}_s, \ell} c_{\tilde{\ell},\ell}\, e^{i\boldsymbol{\vartheta}\cdot\boldsymbol{n}_s}\, |\boldsymbol{m}_c, \boldsymbol{n}_s\,;\,\ell\rangle \tag{41}$$

assuming that the full $d$-dimensional lattice is divided into small pieces whose positions are denoted by $\boldsymbol{m}_c$, while $\boldsymbol{n}_s$ still indicates the relative location of sites in each small piece (It may help to remember that $\boldsymbol{i} \equiv N_s \boldsymbol{m}_c + \boldsymbol{n}_s$ assuming the full lattice is divided into pieces of square or cubic shape). Subsequently $\boldsymbol{\vartheta}$ is the $d$-dimensional quasi-momentum and the coefficients $c_{\tilde{\ell},\ell}$ must be determined such that the corresponding lowest-order energies of PWF $|\boldsymbol{m}_c, \boldsymbol{\vartheta}\,;\,\tilde{\ell}\rangle$ become diagonal with respect to the their orbital indices $\tilde{\ell}$. Then, following the same procedure which led to Eq. (13), the lowest-order energies can be formally written as $\varepsilon_{\tilde{\ell},\boldsymbol{\vartheta}}^{(0)}\big[\boldsymbol{t}(\boldsymbol{m}_c), \boldsymbol{\mu}(\boldsymbol{m}_c)\big]$. The approximate lowest-order energies again become asymptotically exact for infinitely large lattices, owing to the fact that the first-order corrections scale as either $N_s^{-1}$ or $M_c^{-1}$, with $N_s$ and $M_c$ being the total number of sites in each small piece of the lattice and the total number of pieces. We can thus write the formal relation

$$D(\omega) \approx \Re \int d^d x \int d^d \vartheta \sum_{\tilde{\ell}} \delta\Big(\omega - \varepsilon_{\tilde{\ell},\boldsymbol{\vartheta}}^{(0)}\big[\boldsymbol{t}(\boldsymbol{x}), \boldsymbol{\mu}(\boldsymbol{x})\big]\Big), \tag{42}$$

to find the DOS of a general position-dependent multi-band TB model in $d$-dimensions and in the infinite-size limit. In general, the expression (42) cannot be analytically evaluated except for special forms of position-dependent $\boldsymbol{\mu}(\boldsymbol{x})$ and $\boldsymbol{t}(\boldsymbol{x})$. Nevertheless, one can always evaluate the integrals numerically to find the DOS. We should also mention that typically numerical evaluation of integrals such as those in Eq. (42) is not computationally expensive when the size of the matrices becomes very large, as opposed to the direct numerical calculations based on exact diagonalization of the Hamiltonian.

Furthermore, and similar to the 1D single-orbital case discussed in Sec. 5, we can treat the expression as an integral equation that corresponds to the inverse problem of devising the model by knowing its DOS. Therefore, employing the numerical methods for solving integral equations, in principle, we can numerically compute $\boldsymbol{\mu}(\boldsymbol{x})$ and $\boldsymbol{t}(\boldsymbol{x})$ as the solutions of the inverse problem.

# 8    Conclusions

We have considered a general tight-binding model with smooth position-dependent parameters and obtained an expression for its DOS. This expression, which becomes exact in the infinite-size limit, has been derived using a mixed basis interpolating between Bloch and maximally localized Wannier functions. For the one-orbital 1D case, we found the exact solution of the inverse problem, *i.e.* finding the spatial variation of hopping parameter for a given DOS. Then, we constructed a correspondence between position-dependent short-range hopping models and those having position-independent but long-range hoppings. By extension of the framework to the most general (higher dimensional multi-orbital) non-interacting lattice models, we obtained an integral form for the DOS in terms of a general hopping matrix. Our findings provide a concrete method of engineering the DOS by manipulating hopping parameters, which paves the way for a variety of applications.

# A  Solution of the inverse problem

In order to prove the solution of the integral equation in the main text, we first show that it is related to the so-called Abel's equation [28]

$$f(x) = \int_x^{x_0} dy \, \frac{u(y)}{\sqrt{y-x}}. \tag{43}$$

This relation can be easily seen by using $\Omega = \omega^2$, $\tau = 4t^2$, $f(\Omega) = D(\omega)$, and $u(\tau) = \mathcal{Y}(t)/(8t)$, which cause Eq. (19) to transform to

$$f(\Omega) = \int_\Omega^{\Omega_{\mathrm{max}}} d\tau \, \frac{u(\tau)}{\sqrt{\tau-\Omega}}. \tag{44}$$

In order to find the solution of the Abel's equation (43) we can invoke the identity

$$\pi = \int_z^y \frac{dx}{\sqrt{(x-z)(y-x)}} \tag{45}$$

which can be readily checked by direct evaluation of the integral. We then multiply both sides of Eq. (43) with $(x-z)^{-1/2}$ and integrate over $x$ to obtain

$$\int_z^{x_0} dx \, f(x) \, \frac{1}{\sqrt{x-z}} = \int_z^{x_0} dx \, \frac{1}{\sqrt{x-z}} \int_x^{x_0} dy \, \frac{u(y)}{\sqrt{y-x}} \tag{46}$$

$$= \int_z^{x_0} dy \, u(y) \int_z^y dx \, \frac{1}{\sqrt{(x-z)(y-x)}} \tag{47}$$

$$= \pi \int_z^{x_0} dy \, u(y), \tag{48}$$

where in the second line we interchanged the order of integration, and to obtain the third line we used the identity of Eq. (45). Now the final result above indicates that if there exists a solution for $u(y)$, it is given by

$$u(z) = \frac{-1}{\pi} \left( \frac{d}{dz} \right) \int_z^{x_0} dx \, \frac{f(x)}{\sqrt{x-z}}. \tag{49}$$

Again replacing $z = 4t^2$, $x = \omega^2$, and $x_0 = \Omega_{\mathrm{max}}$, we find

$$\frac{\mathcal{Y}(t)}{8t} = \frac{-1}{4\pi} \left( \frac{1}{t} \frac{d}{dt} \right) \int_{4t^2}^{\Omega_{\mathrm{max}}} d(\omega^2) \, \frac{D(\Omega)}{\sqrt{\omega^2 - 4t^2}}, \tag{50}$$

which results in Eq. (21) presented in the text.

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
