# Peer review of "Engineering spectral properties of non-interacting lattice Hamiltonians"

_SciPost Physics_

## Round 1 · Referee Report · Anonymous (Referee 1) · 2021-5-28

Report

The authors derive an integral formula for the density of states (DOS) of a generic noninteracting tight-binding model with position-dependent hoppings t(x) and on-site energies \mu(x). Their focus is on the simpler case of 1D tight-binding models with a single orbital per site, but they also provide a generalization to higher dimensions and multiorbital models. Their approach utilizes a hybrid Bloch-Wannier basis which diagonalizes the Hamiltonian for smooth hoppings in the limit of an infinite lattice. The authors then use their result to solve the inverse problem, namely, how to construct the function t(x) (assuming \mu(x)=0) that gives rise to a given DOS. Finally, they also consider the inverse problem but for translationally invariant tight-binding models with longer-range hopping.

This in an interesting study which employs clever mathematical tricks to solve a rather general problem in condensed matter physics (one could say more generally, in quantum mechanics) that, to my knowledge, has only been treated purely numerically in the prior literature. It should be of interest to a fairly wide audience of researchers in the quantum simulation/designer Hamiltonian communities. The paper is well written, clearly organized, and technically sound as far as I can tell. If the authors can address my questions below, I would be happy to recommend the manuscript for publication.

  1. The integral over position in Eqs. (2) and (18) is reminiscent of taking the trace of the imaginary part of the retarded single-particle Green’s function in the position basis, which is a standard way of computing the DOS. Is it possible to derive those equations using Green’s function methods?

  2. I find the discussion of van Hove singularities on p. 9 a bit confusing. The authors write that “simple 1D TB models typically have van Hove singularities at the band edges as D(\omega)\propto\sqrt{\omega_{max}^2-\omega^2}, yet it remains finite and differentiable, otherwise.” The simplest 1D TB model I can think of, with uniform hopping -t, has van Hove singularities of the inverse square root form at the band edges with \omega_{max}=2t. These are integrable but still divergent. Given the reference to the Wigner semicircle law in the following section, I think the authors may have in mind random TB models, but if so, this should be mentioned explicitly. Also in that same paragraph, when referring to the logarithmic behavior at an ordinary van Hove singularity, it is again not clear if the authors are referring to band-edge or band-center singularities, since both are discussed earlier in the paragraph.

  3. In Sec. 4, the authors use nondegenerate perturbation theory to compute finite-size corrections due to H_1^{intra} and H_1^{inter}. Naively, H_1^{intra} will couple degenerate “Bloch” states at \vartheta=\pi and \vartheta’=-\pi, for example, as in the standard nearly-free-electron problem. Shouldn’t one use degenerate perturbation theory for those terms?

  4. For all the numerical examples of DOS chosen, there is no gap in the spectrum. Does the smoothness requirement for t(x) and \mu(x) necessarily imply a gapless spectrum? For example, the simplest 1D TB model that has a gap is the SSH model, but the function t(x) would not be smooth in this case. Conversely, is it possible to solve the inverse problem for a gapped DOS by the authors’ method?

  • validity: -
  • significance: -
  • originality: -
  • clarity: -
  • formatting: -
  • grammar: -

Author:  Ali G. Moghaddam  on 2021-07-28  [id 1622]

(in reply to Report 1 on 2021-05-28)

Thank you very much for your comment! We have replied to it in the attached pdf.

Attachment:

reply-1.pdf

---

## Round 1 · Referee Report · Anonymous (Referee 2) · 2021-6-4

Strengths

  1. The paper is well-written and very timely.
  2. In general it is in a new area of research with activity just picking up.
  3. The work contains a general idea and then some examples of 1D lattices. 4.Numerically it can be easily implemented.

Weaknesses

  1. The results are valid for 1D lattices which is absolutely fine, whereas it may not be that straight forward for higher dimensions as claimed by the authors.
  2. Some clarifications are necessary.

Report

The authors provide an alternative method to derive the density of states in 1d lattices when the hopping amplitudes and the on-site potentials are position dependent. The method employ a combination of Bloch and Wannier functions by decimating the lattice into pieces (with equal number of sites in each) and then using a quasi-momentum within the small chain (Bloch-type) and localized orbitals for each of the smaller chains (Wannier). Then the Hamiltonian is rewritten using the new decimation and labelling and subsequently the authors carefully do perturbation theory to extract the energy spectrum and the DOS in terms of the hopping parameters that are position dependent, the local chemical potential and their derivatives (as well as the quasi-momentum) .

I find the work very interesting, timely and the paper well-organised and clearly written. It should be published in SciPost after some minor considerations:

  1. It is straightforward to add disorder into a perfectly translational invariant system. This approach can provide some light into such problems. The authors could comment on that as it is an extension within the limits of the theory. Can the same integral as their result of the DOS obtained by real space Green's functions methods?

  2. In 2D there are more recent studies and classifications of Van-Hove singularities and real systems that exhibit this physics, probably the authors should comment on those and connect their work (e.g. PRResearch 2, 013355 (2020), PRB 101, 125120 (2020), PRL 123, 207202 (2019)).

  3. What are the physical requirements for a smooth t(x) and \mu(x)? Can the authors envisage a possible experiment that can test the results? I believe this would strengthen the work considerably.

  • validity: high
  • significance: high
  • originality: high
  • clarity: top
  • formatting: perfect
  • grammar: perfect

Author:  Ali G. Moghaddam  on 2021-07-28  [id 1623]

(in reply to Report 2 on 2021-06-04)
Category:
answer to question

Thank you very much for your comment! Our reply has been attached as a pdf.

Attachment:

reply-2.pdf

---

## Round 1 · Referee Report · Anonymous (Referee 3) · 2021-6-20

Strengths

1- Explicit results with worked examples.

Weaknesses

1- Just single-particle physics. 2- Not properly placed into context.

Report

The authors investigate the spectral properties of lattices with position-dependent hopping amplitudes and on-site potentials. They find the integral expression Eq. (2) for the single-particle density of states (DOS) and discuss the inverse problem of constructing the hopping amplitudes in a lattice model yielding a given DOS. The analytic results are compared to and validated by numerics obtained by exact diagonalization of the hopping model on the lattice.

I have a number of questions and concerns: 1- The first sentence of the Introduction ("The density of states (DOS) is a key physical quantity in condensed matter physics - ...") is too general. The single-particle DOS is central only for single-particle physics, or equivalently non-interacting systems. When interactions are present, even the notion itself would need refining. In particular, the reference to "strongly correlated ... phenomena" at the end of the first paragraph of the Introduction is overstretching things in my opinion. 2- The "PWFs" of Eq. (4) first reminded me of the "projector augmented waves" [P. E. Blöchl, Phys. Rev. B 50, 17953 (1994)] that are famous in the ab-initio context. However, on closer inspection, I have some more basic problems: Eq. (4) appears to define periodic functions. In particular, there is no damping at the boundaries, as suggested by the "partial Wannier " sketch of Fig. 1(a). This raises the question about boundary terms, e.g., in Eq. (6) at the boundaries of ${\cal H}^{\rm intra}$. A related question is: isn't ${\cal B}(x)$ in Eq. (10) and the following line simply proportional to $\delta_{x,0}$? 3- The mapping of a DOS to a hopping model also appears in dynamical mean-field theory (DMFT) when solved with the numerical renormalisation group, see, e.g., Kenneth G. Wilson, Rev. Mod. Phys. 47, 773 (1975) and Ralf Bulla, Theo A. Costi, and Thomas Pruschke, Rev. Mod. Phys. 80, 395 (2008). Consequently, I believe that the essence of chapter 5 of the present manuscript is well-known. The semi-circular density of states Eq. (36) is a classic in DMFT although I have to admit that many related publications glance over the explicit expressions for the related Wilson chain.

Requested changes

The main concerns have been listed in the report. Further details are: 1- Four lines below Eq. (30): a periodic function $\varepsilon(k)$ cannot be "completely monotonic". I believe that this can be remedied by considering different regions of $k$, but the authors should be more precise. 2- Fig. 3 has no panel labels. This is inconsistent with the last paragraph of page 12. 3- Fix excessive lower-casing in titles of references, in particular "I" in [11] and names in [13,17,19,24].

  • validity: ok
  • significance: low
  • originality: low
  • clarity: good
  • formatting: excellent
  • grammar: excellent

Author:  Ali G. Moghaddam  on 2021-07-28  [id 1624]

(in reply to Report 3 on 2021-06-20)
Category:
answer to question
reply to objection

Thank you very much for your comment! Our reply has been attached as a pdf.

Attachment:

reply-3.pdf

---

## Editorial Decision

resubmitted